# Pseudothrombocytopenia—A Review on Causes, Occurrence and Clinical Implications

**DOI:** 10.3390/jcm10040594

**Published:** 2021-02-04

**Authors:** Benjamin Lardinois, Julien Favresse, Bernard Chatelain, Giuseppe Lippi, François Mullier

**Affiliations:** 1Namur Thrombosis and Hemostasis Center (NTHC), CHU UCL Namur, Université Catholique de Louvain, 5530 Yvoir, Belgium; benjamin.lardinois@uclouvain.be (B.L.); j.favresse@labstluc.be (J.F.); bernard.chatelain@gmail.com (B.C.); 2Section of Clinical Biochemistry, University of Verona, 37134 Verona, Italy; giuseppe.lippi@univr.it

**Keywords:** pseudothrombocytopenia, platelets, hematimetry, fluorescence, amikacin, anticoagulants, COVID-19

## Abstract

Pseudothrombocytopenia (PTCP), a relative common finding in clinical laboratories, can lead to diagnostic errors, overtreatment, and further (even invasive) unnecessary testing. Clinical consequences with potential life-threatening events (e.g., unnecessary platelet transfusion, inappropriate treatment including splenectomy or corticosteroids) are still observed when PTCP is not readily detected. The phenomenon is even more complex when occurring with different anticoagulants. In this review we present a case of multi-anticoagulant PTCP, where we studied different parameters including temperature, amikacin supplementation, measurement methods, and type of anticoagulant. Prevalence, clinical risk factors, pre-analytical and analytical factors, along with clinical implications, will be discussed. The detection of an anticoagulant-dependent PTCP does not necessarily imply the presence of specific disorders. Conversely, the incidence of PTCP seems higher in patients receiving low molecular weight heparin, during hospitalization, or in men aged 50 years or older. New analytical technologies, such as fluorescence or optical platelet counting, will be soon overturning traditional algorithms and represent valuable diagnostic aids. A practical laboratory approach, based on current knowledge of PTCP, is finally proposed for overcoming spuriously low platelet counts.

## 1. Introduction

Reported for the first time in 1969 [1], pseudothrombocytopenia (PTCP) has been increasingly described in patients suffering from various disorders and more recently even in patients with coronavirus disease 2019 (COVID-19) [2,3].

This well-known in vitro phenomenon but still underrecognized may lead to misdiagnosis of thrombocytopenia, overtreatment, and further, even invasive, unnecessary testing. This phenomenon can typically be identified by reviewing the peripheral blood smear (PBS), using a different anticoagulant than dipotassium ethylenediaminetetraacetic acid (EDTA) for blood collection, or maintaining the sample at around 37 °C before testing [4]. The prevalence of this phenomenon in EDTA is estimated to be 0.03–0.27% of the general population [5,6,7], though multiple anticoagulant PTCPs with citrate, heparin, or sodium fluoride have also been described [8]. Some parameters can positively influence platelet counting in presence of PTCP. These include the following: (i)In vitro amikacin supplementation can prevent and dissociate platelet clumps due to EDTA, citrate [4], or heparin-PTCP [9] although controversially discussed [10];(ii)Maintaining blood samples to 37 °C could lead to a more accurate platelet count (PC) or prevent EDTA-dependent PTCP [11,12];(iii)A rapid analysis of EDTA blood specimen is advocated, due to time-dependent fall in PC from 1 min after blood collection to 4 h afterwards [4,12,13];(iv)Fluorescence platelet counting can be effective in correcting spurious low PC [14];(v)Capillary blood is prone to platelet clumping [15], but it has been described as less rich in aggregates [16];(vi)Alternative anticoagulants such as magnesium sulfate [5,8,17] and acid citrate dextrose (ACD) [18] are interesting tools to overcome platelets clumps. Hirudin has been studied in few reports only [8,19,20].

Despite the many case studies and the different approaches described, there are few specific expert recommendations on detection and management of PTCP. In this review we discuss the cause, occurrence, and clinical implication of this condition, specifically anticoagulant-dependent PTCP. We start this review by presenting a case study of multi-anticoagulant PTCP, in whom we established a study protocol to assess the PC. A laboratory management algorithm, based on current knowledge, is finally proposed. 

## 2. Case Study

A 47-year-old woman was referred to the local laboratory of Namur Thrombosis and Hemostasis Center (Yvoir, Belgium), for performing a preoperative hemostasis assessment before radical cancer hysterectomy. She did not have any bleeding problems during the delivery of her two children, though her medical history revealed two severe post-operative hemorrhages and menorrhagia. The patient had autoimmune background, as evidenced by high antinuclear antibodies titer (>1:1280) and platelet-bound GPIIbIIIa antibodies. In another laboratory, PTCP with numerous platelet clumps on blood smear had previously been observed using three different anticoagulants: K2-EDTA, sodium citrate, and citrate-theophylline-adenosine-dipyridamole (CTAD). 

According to this information and patient’s multi-anticoagulant PTCP, we established a study protocol to find the most convenient way to assess PC (Figure 1). 

After a 30 min rest, a capillary blood drop from the pulp of index was obtained by a lancing device (Accu-Chek, Roche, Switzerland), which was then used to immediately prepare a blood smear and perform PC. Due to the small volume of sample, the manufacturer’s diluent (DCL-pack, Sysmex Corporation, Kobe, Japan) was used to obtain a blood to diluent ratio of 1:7 for platelet counting in fluorescent (PLT-F), impedance (PLT-I), and optical (PLT-O) channels of a Sysmex XN-1000 blood count analyzer (Sysmex Corporation, Kobe, Japan). Thereafter, a venepuncture in the median cubital vein with 21 gauge needle (Greiner Bio-One, Courtaboeuf, France) was carried out according to recommended venous blood collection practices [21]. Five vacuum blood tubes containing K2-EDTA, five containing 109 mmol/L buffered sodium citrate with blood to anticoagulant ratio of 9:1 (Vacuette^®^, Greiner Bio-One, Courtaboeuf, France), and five containing hirudin 525 ATU/mL (Monovette^®^, Sarstedt, Nümbrecht, Germany) were collected. Each set of anticoagulant tubes was used to study the effect of temperature and in vitro amikacin supplementation (5 mg/mL) [22] on PC at 0.5, 60, 120, 180, and 240 min after blood collection. No significant clinical changes were considered when PC remained over time within the reference change value (RCV) calculated for each channel (14% for PLT-F and PLT-O, 25% for PLT-I) according to the following formula: RCV = 2^1/2^ * Z * (CV_A_^2^ + CV_I_^2^)^1/2^ [23] assuming Z-score of 1.96 used (95% probability) and CV_I_ of 5.6% obtained from the European Federation of Clinical Chemistry and Laboratory Medicine biological variation database [24]. The analysis of blood smear was performed, with special focus on peripheral film (as recommended) [25], and platelet clump was defined as a minimum of five attached platelets [4].

Amikacin-free specimens were diluted with manufacturer’s diluent, for achieving the same dilution as in tubes containing amikacin; the dilution effect of sodium citrate, amikacin, or diluent were calculated. A patient control sample without amikacin or diluent was maintained at room temperature (RT) for each set. PC and platelet surface glycoproteins expression levels by flow cytometry (FCM), platelet function, fibrinolysis, fibrin formation, mixing studies, and serum anti-platelet antibodies were also assessed. 

Platelet clumping was observed irrespective of the types of tube used and until the last measurement at 240 min (Figure 2). PCs according to different anticoagulants, conditions, and measuring channels on XN-1000 hematology analyzer are shown in Figure 3. 

The PC obtained with capillary blood in fluorescent and optical channels were 214 × 10^9^/L and 141 × 10^9^/L, respectively. The impedance method generated no data. In venous blood, the PCs with the PLT-F channel at baseline were approximately 150 × 10^9^/L regardless of type of tube and storing condition. The PLT-O showed higher variability than PLT-F, both at baseline and at different time points. The impedance method generated low PC for each set. Mean platelet volume (MPV), platelet larger cell ratio, and platelet distribution width obtained at baseline in patient control EDTA sample were 13.3 fL, 54.0%, and 18.7%, respectively.

Tubes containing citrate, kept at 37 °C, displayed the greatest stability, independent of amikacin, as did K2-EDTA stored at RT, with the entire range of values remaining within the RCV. Maintenance of EDTA tubes at 37 °C rather than at RT had negative impact on PC, with all analytical techniques used. This was particularly observed for amikacin-free specimens at 37 °C, with many clumps observed on the blood smear. Hirudin specimens displayed the worst stability over time, in all conditions. 

The PC obtained with the reference method in FCM on K2-EDTA (145 × 10^9^/L) was quite similar to that obtained at baseline in fluorescence and optical channels. Platelet surface glycoproteins, platelet function, fibrinolysis, and fibrin formation were normal. In addition, mixing studies and serum anti-platelet antibody test were reported as normal. A Von Willebrand disease (VWD), including type 2B VWD and platelet type VWD was also ruled out. Thus we concluded that there was multi-anticoagulant PTCP and the actual PC was 145 × 10^9^/L.

This single person experimental approach had some limitations. First, control measurements with blood from individuals without PTCP were not carried out. This would have assessed the in vitro cell shrinking or swelling, thus resulting in additionally measurable platelets, observed in the EDTA control tube by the increase of PLT-I and MPV from 13.3 to 15.1 fL within the first hour. This phenomenon does not interfere with PLT-F or PLT-O. Secondly, no control tube at 37 °C was obtained; therefore, a general effect of temperature on platelet clumps could not be excluded. Thirdly, no data were available to explain the decreasing PC in citrated sample, though the hypothesis of calcium release from reversible citrate binding could be suggested. Fourthly, other anticoagulants, including magnesium sulfate, could have been studied, but these were not available in our laboratory when the case study was carried out.

## 3. Mechanisms, Prevalence and Risk Factors

### 3.1. Mechanisms

First described by Gowland et al. in 1969 in a patient with a malignant non-Hodgkin’s lymphoma [1], PTCP was rapidly identified as being in vitro phenomenon with many further observations in healthy subjects, without a significant bleeding phenotype. EDTA-dependent PTCP has also been described in animals such as cats, minipigs, dogs, and horses [26,27,28,29,30].

Manthorpe et al. in 1981 first reported anti-platelet antibody activity based on immunofluorescence experiments using Fab’2-fragments of isolated IgG [31]. Then, Pegels et al. confirmed the existence of an immunologically-mediated phenomenon caused by presence of IgG, IgM, or IgA directed against platelet antigens, manifesting mostly at 0–4 °C in EDTA samples [32]. Other immunological studies proved the varied origin of antibodies, belonging to different classes of immunoglobulins, mainly IgG (and especially IgG1), sometimes concomitantly with IgA and IgM. These could be either acquired or naturally occurring autoantibodies [33,34,35,36]. Bizzaro et al. identified the presence of anticardiolipin antibodies in up to two-thirds of samples containing antiplatelet antibodies [37]. However, a recent study showed that there was no significant difference in anticardiolipin antibody titers between individuals with PTCP and a control group of healthy patients [38]. Unlike these previous observations, PTCP in a patient with clinical anti-phospholipid syndrome was only reported in 2018 for the first time [39]. Cold agglutinins and antinuclear antibodies were found to be significantly more frequent in individuals with PTCP compared to healthy volunteers [38,40,41,42]. To the best of our knowledge, no other immunological studies have been carried out since the beginning of this century. 

Although the phenomenon has not been fully elucidated, it has been suggested that these autoantibodies may be directed against cryptic exposed epitopes of the GpIIbIIIa complex (fibrinogen receptor), a calcium dependent heterodimer, and/or negatively charged phospholipids, therefore responsible for in vitro platelet agglutination after activation via tyrosine kinase [4,12]. It could hence be considered as aggregation rather than agglutination. Interestingly, platelets from patients with Glanzmann’s thrombasthenia (absence of or minimal residual amounts of GPIIbIIIa or abnormal fibrinogen receptor) fail to bind these autoantibodies [42]. Recently, an IgM class agglutinin against collagen receptor GPVI and triggering platelet activation has been identified in citrate anticoagulated blood of a patient suffering from moderate bleeding diathesis [43]. These cryptic epitopes are mostly exposed when platelets react with EDTA [44,45], which irreversibly chelates calcium [46], leading to remarkable conformational changes of these neoantigens [37,47]. This phenomenon may be observed with any EDTA formulation [12], and has also been described for molecules resembling EDTA such as ethylenetriamine tetraacetic acid [48]. Interestingly, this conformational change is not observed with alternative anticoagulants such as citrate [10], although a recent report suggested that conformational changes could also appear with this latter anticoagulant or even with heparin [49]. Other immune complexes interacting with Fc receptors on platelet surface could be implicated in the PTCP process, with varying degrees of specificity [7]. 

The PTCP spectrum may involve other additives, thus describing a new, rarer entity, which can be conventionally called multi-anticoagulant PTCP [3,8,49,50,51,52]. It has even led some authors to mention an “Anticoagulant-induced” PTCP rather than “anticoagulant-dependent” PTCP.

This phenomenon can be persistent or transient, sometimes showing alternating periods without aggregates generation [10]. A recent case of transient PTCP has been described in a COVID-19 patient during viral immunization [2].

Transplacental transmission of maternal IgG leading to PTCP in newborns has been described [53,54,55]. This condition should be considered in neonates with prolonged thrombocytopenia [56]. Pediatrics is therefore not safe from this phenomenon, though the majority of cases were children suffering with infectious diseases [57,58,59,60,61]. Nonetheless, a case of healthy children has been described [62].

Spurious thrombocytopenia could also be related to platelet satellitism around white blood cells. Classically observed as an in vitro phenomenon with platelet rosetting around the cytoplasm of neutrophils, it has been less frequently observed around lymphocytes [7,63]. The underlying mechanism has not been fully elucidated so far, though IgG autoantibodies targeting GpIIbIIIa would be involved in binding to neutrophil Fc Gamma receptor [64]. Other non-immune hypotheses pinpoint that proteins from alpha granules or thrombospondin would be expressed on platelet membrane, thus leading to adhesion to neutrophils [65]. This in vitro process would then evolve towards a more generalized agglutination of platelets and neutrophils in large aggregates containing over 100 cells. The latter phenomenon can be more rarely detected, and seems to be especially EDTA-dependent.

### 3.2. Prevalence 

Several studies, most retrospective, have been conducted for assessing the prevalence of PTCP in different populations. The prevalence in K2-EDTA has been estimated at around 0.03–0.27% in an outpatient population [6,7,35,66,67,68,69,70,71,72], increasing to 15.3% in patients referred to an outpatient clinic for isolated thrombocytopenia [73,74]. Up to 17% of patients with EDTA-dependent PTCP display also a low PC in citrate samples [35]. Three studies evaluated the prevalence of PTCP in blood and platelet apheresis donors, with frequency ranging from 0.01% to 0.2% [75,76,77]. Although the prevalence of EDTA-PTCP did not appear to be significantly influenced by age or gender [12], a recent retrospective Chinese study of 190,940 individuals with regular hospital checkups showed that this condition was significantly higher in males aged 50 years or older [72]. Platelet satellitism is even rarer, approximately 1 every 12,000 blood counts [7,78].

### 3.3. Clinical Risk Factors

Since the discovery in 1969, numerous hypotheses on the origin of anti-platelet antibody production have been formulated, and many studies have attempted to identify some putative risk factors. 

The commonly accepted hypothesis entails antibody production due to cross-reactivity, as recently described in two COVID-19 patients during viral immunization [2,3], or in autoimmune antibodies as described here. The presence of platelet-bound GpIIbIIIa antibodies, found in 80% of cases (as well as in our patient), reinforced this hypothesis.

Figure 4 summarizes the relationship between PTCP and clinical conditions. No particular disease is strongly associated with presence of PTCP, or show significant differences from a control population of healthy individuals. However, the incidence of EDTA-PTCP seems increased with hospitalization or in patients with some specific disorders, such as autoimmune diseases. Infection, pregnancy, and use of medication such as low-molecular-weight heparin (LMWH) have also been identified as significant risk factors [12,38,41]. As earlier mentioned, males >50 years could be at higher risk [72]. Additional conditions found to be especially associated with PTCP are reported in the scientific literature as case reports. These basically include viral infections [3,59,79], sepsis [80], thrombotic and cardiovascular disorder, heparin-induced thrombocytopenia (HIT) [41,81,82], auto-immune disorders [39,83], liver diseases [51,84], cancer [85,86], surgical settings [87,88,89], post stem cell transplantation [90,91], treatment with valproic acid [51], insulin, antibiotics [92], or chemotherapeutic agents such as sunitinib [52]. Some authors prefer to use the term “coincidental PTCP” or “concomitant PTCP” rather than expressly associating this condition with a particular disease [43,81]. Several recent cases with therapy based on monoclonal antibodies have also been described, encompassing GpIIbIIIa [93] or immune checkpoint inhibitor such as pembrolizumab [94]. 

Notably, the identification of PTCP does not seem to enhance the risk of developing a certain disorder [43]. Some individuals have persistent PTCP for decades, without reporting any relevant pathological state. More data are hence necessary to evaluate PTCP as a clinical condition for possible diseases. However, it has already been suggested that it may be advisable to investigate the presence of an autoimmune disease when observing PTCP [38]. In addition, platelet satellitism has been found both in healthy people and in patients with cancer, infections, or autoimmune disorders [78,95]. Critical recognition of PTCP in situation at higher bleeding risk, including patients undergoing cardiac surgery [89,96,97] or with life-threatening disorders such as disseminated intravascular coagulation (DIC) and HIT [12], could prove advantageous. PTCP shall also be accurately ruled out in patients undergoing treatment with GPIIbIIIa antagonists, due to the high risk of stent thrombosis that would result from discontinuation of such therapy [98]. 

## 4. Pre-Analytical and Analytical Influencing Factors

### 4.1. Collection 

Laborious venipuncture, overfilled blood specimens, or blood draws from indwelling lines may trigger platelet clumps by spurious activation of coagulation in vitro [7]. Capillary venous blood seems also more vulnerable to platelet clumping [15], though it has been described as having a lower number of aggregates [16]. 

Although capillary blood displayed the highest PC in our case report, uncertainty remains concerning buffer dilution and a possible matrix effect, since the analyzer is calibrated for using venous blood. 

### 4.2. Aminoglycosides and Other Compounds

Aminoglycoside supplementation in EDTA blood sample has been extensively investigated [5,9,13,50,69,99,100], maybe for its wider hospital availability compared to other compounds.

In our case, the presence of a stabilizer in the preparation of the aminoglycoside (e.g., sodium citrate) [101] may explain why the addition of amikacin in citrated tubes did not resolve the issue. In contrast, K2-EDTA supplemented with amikacin displayed greater stability of platelet values, thus revealing a possible benefit from dual anticoagulation. However, the package insert does not specify the concentration of sodium citrate in solution [22], so that the role of stabilizers in PTCP needs to be further elucidated. 

More importantly, amikacin is not always effective in correcting PCs in vitro [13], since its activity is dependent on the type of anticoagulants to which it is added [9]. The mechanism of aminoglycosides activity in PTCP remains hypothetical [9,11], and its use controversial [10].

Other aminosides including kanamycin can be added in EDTA-samples to prevent or even dissociate platelet aggregates, though displaying variable outcomes [13,50,69]. Supplementation with other pharmaceutical drugs in blood samples such as potassium azide, calcium chloride, antiplatelet agents including GpIIbIIIa inhibitors, thromboxane inhibitors, irreversible cyclooxygenase inhibitors, or apyrase have also been described [4,12]. However, no single additive has been shown to be reproducibly effective against platelet clumping. Moreover, these laboratory procedures are time-consuming, and must be hence reserved to specific cases. 

### 4.3. Temperature 

Alternative strategies, such as maintaining the whole blood sample at 37 °C until analysis, can be seen as reliable means for preventing platelet clumping, though will not be effective to solve the interference when already present, as in our case. Interestingly, the reticulocyte channels for PLT-O and PLT-F on Sysmex XN^®^ instrumentation are pre-warmed before platelet enumeration, which has plausibly influenced the counting. Future evaluations should also consider comparing PC at lower temperatures, given the observation of higher PCs obtained at RT than at 37 °C in K2-EDTA specimens in our case. 

Indeed, heating EDTA-blood samples to 37 °C does not guarantee an accurate PC in 20–35% of reported cases [11,35,99], either because this phenomenon is sometimes temperature-independent, or some autoantibodies (especially IgM) have best clumping activity at 37 °C, thus resulting in instant aggregation in any anticoagulated blood sample [10,102]. When performed, incubation at 37 °C should be initiated as early as possible after phlebotomy [103].

The broad spectrum of described case reports, even for similar conditions (e.g., anticoagulants, temperature) is perhaps attributable to different physico-chemical properties of the antibodies, as well as to confounding variables (time after sampling, pH, medications, and so forth).

### 4.4. Time

The need for rapid analysis of an EDTA blood tube has been suggested by French guidelines [4] and others [10]. This is due to the time-dependent fall in PC, from 1 min after blood collection to 4 h afterwards [12,13,101,104,105]. Citrate-anticoagulated specimen also evidence underestimated PC when analyzed over 3 h after phlebotomy [106]. This may be explained by the reversible calcium-chelation, which triggers progressive aggregates generation. However, time was not a substantial parameter in our case report, as the PC remained stable in K2-EDTA stored at RT. 

### 4.5. Analytical Techniques 

The detection of PTCP is highly dependent on the analytical technique, each showing its own sensitivity and/or vulnerability to the presence of platelet aggregates. 

The analytical technique has been a cornerstone for partially avoiding interference due to clumps for this patient. Indeed, PLT-F, PLT-O, and FCM displayed the highest PC, thus suggesting partial dissociation of aggregates using these techniques, as opposed to the impedance method, which was the most vulnerable to platelet clumping. This dissociation effect, independent of manufacturer’s fluorescent dye staining, has been assumed in a recent Chinese study [14]. However, sample treatment with hematology or flow cytometry analyzer reagents could not be carried out in our case.

The majority of clinical laboratories are currently using impedance technique for platelet counting, as a consequence of larger availability of instruments based on this standard technology in the market. This method, encompassing variation in electric current intensity when a blood particle passes through two electrodes, does not discriminate platelets from other blood elements with similar same size range [4,107,108]. This technique hence displays high inaccuracy in several clinical situations, despite implementation of computerized algorithms (“moving threshold” and “curve fitting”) [107], and remains vulnerable to platelet clumps, as highlighted in this case report. Depending on size, platelet clumps are then enumerated as single large platelets, or as small lymphocytes, resulting in spuriously low PC, and sometimes leading to falsely elevated leukocyte count [4]. Interestingly, the impedance histogram is a useful tool to detect PTCP, as it may display sawtooth irregularities in the curve and serrated tail, sometimes with no return to the baseline at 20 fL, along with the inability of the analyzer to determine a cut-off between platelets and red blood cell fragments or microcytic red blood cells [4,14,108,109].

Under these measurement conditions, the presence of large platelets (e.g., macroplatelets or giant platelets) as observed in constitutional thrombocytopenias, myeloproliferative neoplasms, or immune thrombocytopenias, may also lead to underestimation of PC, whether or not in vivo thrombocytopenia is present. This less known phenomenon could hence lead to PTCP, not related to the EDTA, but to limitations of current instrumentation when analyzing large platelets.

The impedance method is usually used in routine settings for PC, though laboratory technicians could switch to optical-based or fluorescence-based methods using the reticulocyte mode on demand [110,111]. In our laboratory, these methods are set up as “reflex test”, when abnormal histograms of RBC or platelet size are encountered.

The optical platelet-counting method identifies platelets with laser light scatter technique. It uses bi-parametric analysis, based on size and RNA content of particles (except for ADVIA counters). This method overcomes some of impedance drawbacks [107], though obtaining an accurate PC remains challenging in some cases. In our case, PLT-O at baseline showed PC close to FCM reference method. 

The fluorescence-based technology is currently marketed by two manufacturers, Sysmex (Kobe, Japan) and Mindray (Shenzhen, China), respectively. On Sysmex XE and XN instruments, this technique combines scattered light and side fluorescence detectors. A fluorescent dye (oxazine) is used beforehand to stain platelets and reticulocytes. It has been shown as a reliable method for accurate platelet counting in thrombocytopenic patients, and one of the most reliable for taking clinical decisions on platelet concentrate transfusions [112,113,114,115,116,117]. Compared to PLT-O, PLT-F can more clearly distinguish PLT from other blood cells, and is also more accurate for analysis of giant thrombocytes [4,115]. More importantly, it has recently been described as effective in correcting spurious low PC on the two platforms Sysmex XN 9000 and Mindray SF-cube, even suggesting that an alternative anticoagulant will probably no longer be necessary in case of EDTA-PTCP [14,108]. Limitations of this enumeration technique are longer turnaround time, increased reagent costs due fluorescent dye, and need of larger sample volume [113]. Additional studies with large number of patients are still needed to demonstrate that Sysmex XN 9000 and Mindray SF-cube are actually correcting spuriously low platelet counts.

Although the immuno-platelet counting using monoclonal CD41 and CD61 antibodies by FCM is the reference assay for accurate PC [118], this approach requires specific skills, is time consuming, and not applicable in practice when dealing with PTCP. It has been poorly investigated in case of PTCP, and our results show that PLT-F and PLT-O were quite similar. This led us to suggest that even this method does not allow optimal thrombocyte counting in presence of PTCP, despite a possible partial aggregates dissociation. 

Based on the available literature and this report, we hence suggest fluorescence-based counting as the most useful second-line technique given the nature of FCM and the larger variability observed with PLT-O. 

Alternative microscopic methods in case of multi-anticoagulant PTCP should also be considered, such as manual counting on native whole blood sample [4] (i.e., capillary blood or venous blood from discard tube) or on PBS, the latter being able to benefit from double platform counting [119] by multiplying the platelet to red blood cell ratio by the concentration of red blood cells counted on the analyzer. The accuracy and efficiency of this counting method can be improved by using digital microscopy. However, this has been shown on normal samples but has never been evaluated in the presence of platelet clumps. 

### 4.6. Alternative Anticoagulants 

Sodium citrate is the most widely used alternative anticoagulant to EDTA due to its wider availability on the market and its lower risk of clump formation. However, PTCP has been described with EDTA and citrate concomitantly, and even with more than two anticoagulants [3,8,49,50,51,52], including in this case report. The major disadvantage of citrate is the introduction of a dilution factor, since the anticoagulant is only available in liquid form. The most commonly used correction of 110% may not be sufficient, because the PC is underestimated compared to EDTA [106]. Two reports showed the need for a higher corrective factor; one suggested 117% and the other 125%, respectively [11,120]. However, French guidelines still recommend the 110% correction factor, and caution is advised for citrate samples processed more than 3 h after collection. The PC on citrate therefore remains slightly insufficient when correcting EDTA-PTCP, albeit to a lower extent than in the presence of aggregates.

Heparin is another commonly used alternative anticoagulant, though is currently no longer recommended by the French guidelines. Its weak staining properties make interpretation of blood smears challenging due to the presence of halos around cellular elements, while it seems also not always efficient to prevent PTCP [4]. 

Sodium fluoride [109,121], CPT (Trisodium citrate, pyridoxal-5′ phosphate and Tris) [122,123] and CTAD (citrate, theophylline, adenosine, and dipyridamole) can also be used. No cases of PTCP have been reported for the last two additives. Although the latter also suffers from a dilutional effect due to anticoagulant availability in liquid form, several studies including some on animals showed its benefit in resolving EDTA-PTCP [124,125,126].

Two anticoagulants especially useful in PTCP should also be considered, although they have not been studied here. One of these, based on magnesium sulfate, is effective to prevent PTCP and hence recommended for PC once PTCP has been documented [10,17,101,127,128]. Magnesium inhibits platelet aggregation through its ability to prevent intracellular calcium influx and by inhibiting thromboxane A2 formation. In addition, it interferes with fibrinogen binding to stimulated thrombocytes [17,101]. Historically used for platelet counting, it has been offset by automation in hematology, which has propelled EDTA as the reference additive. Its effectiveness has even been proven in multi-anticoagulant PTCP, as the most effective compound to prevent platelet clumps among the five anticoagulants [8]. No case report of PTCP with this additive has yet been published to the best of our knowledge. The other additive, acid citrate dextrose (ACD), may almost invariably prevent platelet aggregation [18], even more effectively than using sodium citrate, due to the more acidic content along with a lower aggregation. Before being replaced by EDTA, the ACD was the most widely used additive for routine platelet enumeration [31]. 

Given the few hirudin-PTCP reports described and its increasing use in clinical laboratories, especially for multiple electrode aggregometry [129], the hirudin tube was also studied in this experimental protocol. This study can be added to the previous reports that showed that hirudin specimens had no advantage in correcting PTCP [8,19,20], in addition to being expensive.

## 5. Clinical Implications

Although laboratory awareness of this PTCP has led to a proactive attitude towards clinicians, this phenomenon continues to have clinical consequences. Recent published cases of PTCP, leading to undesirable clinical implications, are summarized in Table 1.

The PTCP may have only minor pathophysiologic significance. However, this situation must be distinguished from true in vivo platelet clumps detected by chance during a blood test. Some reports have mentioned this possibility. One described in vivo platelets clumps due to arterial occlusive stent [3]. The pathogenesis of clump formation in arterial thrombosis has no strict relationship with the dynamics of aggregates formation in PTCP. Other cases described in vivo platelet clumping in patients with type 2B VWD, who were misdiagnosed with EDTA-PTCP [131,132,133]. Interestingly, morphological features of clumps in this setting are different from the aggregates encountered in EDTA-dependent PTCP, being characterized by more heterogeneous and larger sizes [15,132]. Findings of platelet clumps on blood smears from whole blood (either finger prick or discard tubes) may suggest this pathology [131]. Although we did observe platelet clumps on whole blood samples, VWD including VWD 2B and platelet type VWD could be ruled out in our patient. A significant increasing of in vivo platelet clumps have also been described in healthy marathon runners, 30 min after the race [134]. 

As the in vitro phenomenon of PTCP could be misdiagnosed with thrombocytopenia, it does affect diagnostic, management, and therapeutic decisions. Inadequate treatments such as platelet transfusion [2,80,128,130,135] or high doses corticosteroids against immune thrombocytopenia [49,83,100] may be frequently used, hopefully with minor iatrogenic consequences. Nonetheless, other implications such as discontinuation of heparin therapy could be potentially life-threatening [81]. Detection of PTCP can even modify the decision to initiate heparin therapy, especially when managing acute coronary syndrome [96]. Unnecessary investigations such as a bone marrow biopsy and unwarranted surgical procedures such as splenectomy have also been described [7,96,136,137]. 

Given the potential life-threatening situations, some authors suggested that PTCP must be in the differential diagnosis, or even ruled out, before considering a diagnosis, for example in patients with cancer [94] or HIT [81]. Interestingly, a study observed that 88% of patients with EDTA and citrate dependent-PTCP had anti-platelet factor 4 (PF4)/heparin antibodies [82], thus potentially leading to diagnose HIT and discontinuing heparin therapy. However, larger clinical studies are needed to demonstrate the presence of these antibodies in PTCP [41]. In addition, laboratory tests used for HIT diagnosis must be integrated into a clinico-pathological approach. In a patient with positive pre-test score, the presence of anti-platelet factor 4 (PF4)/heparin antibodies should be confirmed by a functional assay, which measures heparin-dependent platelet activation/aggregation induced by HIT antibodies [138]. Finally, a B-cell lymphoproliferative disorder could be suspected in the presence of platelet satellitism around lymphocytes [63]. 

## 6. Management in the Laboratory and Practical Flowchart

A careful search in the literature about management of PTCP shows different approaches, though most of these generally follow a three step process: identification, confirmation, and prevention. Below, we propose a specific algorithm, based on the current knowledge of PTCP (Figure 5). This flowchart is based on the recent decision tree from the last recommendations of Groupe Francophone d’Hématologie Cellulaire (GFHC) for thrombocytopenia [4]. 

Once the presence of clots has been ruled out, inspection of histograms and scattergrams of RBCs, platelets, and white blood cells is carried out during the staining of the May–Grünwald–Giemsa smear. These may already suggest the presence of platelet satellitism, giant platelets, or aggregates, which must then be confirmed on PBS. If fluorescence or optical measurements are available, they should be considered either on the basis of histograms or on PBS. When these methods do not remove the interference, an alternative anticoagulant should be considered to differentiate between a preanalytical problem, an EDTA-dependent PTCP, or a multi-anticoagulant PTCP. In the latter case, third-line anticoagulants such as magnesium sulfate or ACD are recommended. Interestingly, these anticoagulants are already used as second-line anticoagulants instead of citrate in some laboratories. Manual counting or variation of physico-chemical parameters should be performed in a limited number of cases. Finally, the preferred counting technique should always be reported, at least within the laboratory information system (LIS), thus limiting further investigations and potential clinical repercussions.

## 7. Conclusions

PTCP is a complex phenomenon, influenced by the method used for counting thrombocytes and including also preanalytical issues. It seems that no particular disease may be specially associated with PTCP. This condition could still lead to misdiagnosis of thrombocytopenia, impairing diagnosis, management, and therapeutic decisions. Most cases of EDTA-dependent PTCP can be corrected by using different anticoagulants, whilst multiple anticoagulants PTCP is a less acknowledged laboratory phenomenon, resulting in more analytical challenges. Observing PTCP in individuals does not increase the risk of developing future disorder. Conversely, the incidence of PTCP is higher in patients with identified risk factors, including male sex, age over 50 years, underlying diseases, or therapy with drugs such as LMWH. New technologies such as fluorescence or optical platelet counting should be implemented in clinical laboratories, as they will provide a valuable and suitable support for correcting spuriously low PC. Heating the whole blood specimen at 37 °C, in vitro amikacin supplementation, or rapid sample analysis are more laborious strategies, which are not always effective to improve the accuracy of PC, as observed in this case report. Alternative anticoagulants to EDTA (e.g., magnesium sulfate or ACD) should now be reconsidered in sample with clear evidence of PTCP. Finally, the counting method should always be reported, at least in the LIS. 

## Figures and Tables

**Figure 1 jcm-10-00594-f001:**
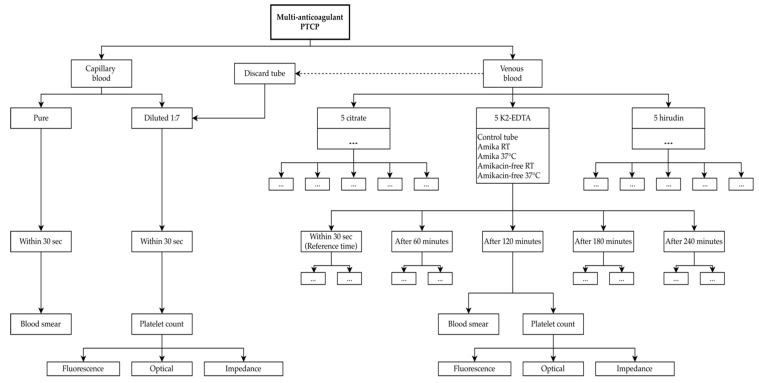
Study protocol. Empty boxes just repeat the same elements as adjacent boxes. The time is expressed in seconds or minutes after blood collection. Abbreviations: PTCP, Pseudothrombocytopenia; RT, room temperature; EDTA, ethylenediaminetetraacetic acid.

**Figure 2 jcm-10-00594-f002:**
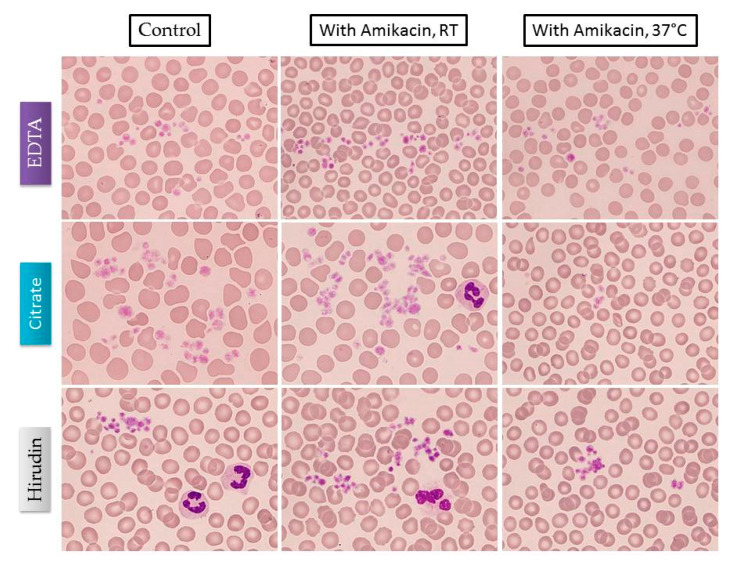
Platelet clumps observed on blood smears from samples obtained at baseline according to different anticoagulants tubes and conditions. Clumps were observed until the last observation at 240 min. Although not illustrated, amikacin-free specimens also displayed several clumps. Abbreviations: EDTA, ethylenediaminetetraacetic acid.; RT, room temperature.

**Figure 3 jcm-10-00594-f003:**
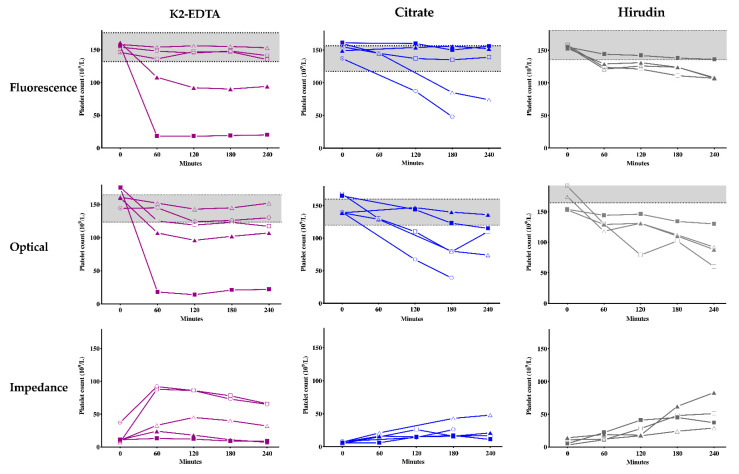
Changes of platelet counts over time according to different anticoagulants (K_2_-EDTA, Na-citrate, hirudin), measurement methods and conditions (control, amikacin or not, room temperature or 37 °C). The grey zone corresponds to the RCV based on the control tube at baseline for K2-EDTA, sodium citrate and on amikacin-free tube at room temperature for hirudin. Control hirudin tube was excluded since a blood clot was found. 🞅 Control, RT, △ Amikacin, RT, ▲ Amikacin, 37 °C, ☐ No amikacin, RT, ■ No amikacin, 37 °C.

**Figure 4 jcm-10-00594-f004:**
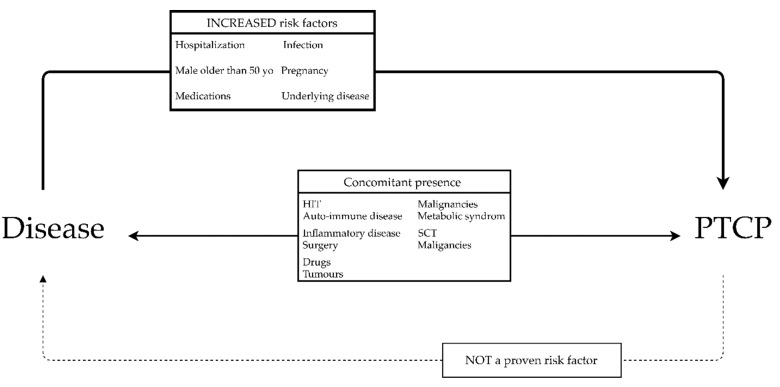
Relationship between anticoagulant-dependent pseudothrombocytopenia and disease. Increased prevalence of PTCP has been described in populations in the upper box. Concomitant presence of PTCP and a disease have been observed without proving an increased risk for one or the other. The presence of a PTCP does not increase the risk of developing a specific disorder. Abbreviations: HIT, heparin-induced thrombocytopenia; LMWH, low-molecular-weight heparin; SCT, stem cell transplantation.

**Figure 5 jcm-10-00594-f005:**
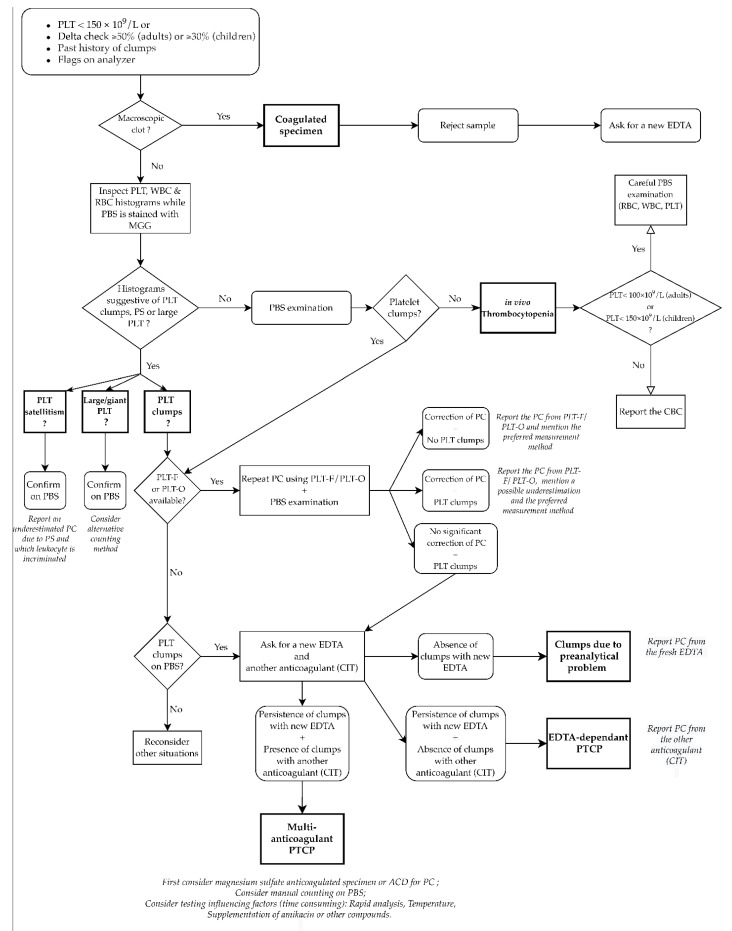
Modified flowchart diagram of EDTA thrombocytopenia based on the decision tree from Baccini et al., 2020 (GFHC, French Society of Haematology). This provides a practical approach for laboratory staff in the presence of thrombocytopenia. ACD: acid citrate dextrose; CBC: complete blood count; CIT: citrate; MGG: May–Grünwald–Giemsa staining; PBS: peripheral blood smear; PC: platelet count; PLT: platelet; PS: platelet satellitism; PLT-O: optical-based platelet count; PLT-F: fluorescence-based platelet count; PTCP: pseudothrombocytopenia; RBC: red blood cells; WBC: white blood cells.

**Table 1 jcm-10-00594-t001:** PTCPs with clinical impact reported in the literature since 2015.

Author (Ref.)	Anticoagulant-Induced Ptcp	Medical Condition	Clinical Implication	Confirmation on Alternative Anticoagulants
Akyol et al., 2015 [130]	EDTA	SLE & Lupus nephritis	Unnecessary platelet transfusion	Citrate
Kohlschein et al., 2015 [128]	EDTA, Citrate	Paroxysmal atrial fibrillation	Delayed cardiological intervention	Magnesium sulfate
Greinacher et al., 2016 [98]	EDTA, Citrate	ACS under GpIIb/IIIa antagonist (Eptifibatide)	Therapeutic management issues and wrongly transferred in ICU	ND
Shi et al., 2017 [80]	EDTA	Sepsis	Unnecessary platelet transfusion	Citrate
Li et al., 2020 [2]	EDTA	Viral infection (COVID 19)	Unnecessary platelet transfusion	Citrate
Kuhlman et al., 2020 [3]	EDTA, Citrate, Heparin	Viral infection (COVID 19)	Associated with an arterial occlusive stent (STEMI)	None
Zhong et al., 2020 [49]	EDTA, Citrate, Heparin	Viral infection (gastroenteritis)	Treated with dexamethasone due to misdiagnosis of ITP	None

Abbreviations: ACS, Acute coronary syndrome; ICU, Intensive care unit; ITP, immune thrombocytopenia; ND, not determined; SLE, Systemic lupus erythematosus; STEMI, ST-Elevation myocardial infarction.

## Data Availability

Please contact the corresponding author.

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
