# Peer review of "Pseudothrombocytopenia—A Review on Causes, Occurrence and Clinical Implications"

_jcm, 2021, doi:10.3390/jcm10040594_

Round 1

Reviewer 1 Report

The presented case of „multi-anticoagulant“related PTCP is expended through a comprehensive review of the current literature. The review with the exception of a few points is appropriate to today's knowledge.

The case report is supplemented by a demanding experimental part with the aim to demonstrate how the real platelet count might be determined or at least estimated. In this context, some major critical comments and suggestions for changes are necessary:

Major criticism:

Figure 3 represents the results of a very ambitious experimental design. The figure is very detailed and consists of at least 9 sub-figures with platelet counts (PCs) measured by different techniques at different times, anticoagulated by different anticoagulants and after adding additives such as amikacin.
The presentation of numerous data in one figure is not self-explaining and needs modification: The hirudin data do not substantially contribute to the central results or conclusions of the paper and can be summarized in the text and the respective graphs might be omitted. This reduction allows enlarging the remaining graphs and symbols.

How can you prove that the higher PCs as measured by Plt-F and Plt-O result from dissolution of aggregates? It would be therefore of interest to treat a patient´s sample with the respective reagents used for PC measurement and to prepare control smears for aggregate examination.
Despite the matrix and calibration problems, the capillary platelet counts (PCs) give an important hint for estimating the probable PC of the patient. To approximate the PC by microscopic counting would have been another approach as proposed by the authors under 4.5. Although FMC is the standard for PC, platelet aggregates may also influence the PC. For the proof of whether Plt-F and Plt-O measure the correct platelet number not affected by aggregates, the real (true) PC is decisive.

The low PC using K2-EDTA and Plt-I is certainly due to aggregates but also to cell shrinking by reagents and diluents. This assumption explains the increase of PC within the first 60 minutes of storage: K2-EDTA leads to platelet swelling resulting in a proportion of additionally measurable platelets. Therefore it is absolutely necessary to include data of the mean platelet volume (MPV, P-LCR and PDW) of the patient despite probable interferences by the aggregates. Cell shrinking or swelling does not interfere with Plt-F or Plt-O.
These relationships could be substantiated much better if the authors had carried out control measurements with blood from individuals without PTCP.

How do the authors explain the low counts in the EDTA-anticoagulated samples without Amikacin when kept at 37 0C and why is this totally different in citrate-anticoagulated blood?
What is the reason that the addition of Amikacin obviously leads to better PCs?
The authors postulate that the decreasing PC in the citrated sample is due to calcium release from reversible citrate binding. Is there any proof e.g. smears from parallel experiments?

Last not least it is a shortcoming that the interesting results were only obtained from one person in a single experimental approach.
Some questions could possibly have been answered by replacing hirudin by another anticoagulant e.g. magnesium sulfate.

Minor points of criticism:

“PTCP patients”? Most candidates are “patients” but not because of PTCP.
“Individuals with PTCP” is more correct. PTCP is a laboratory in vitro phenomenon.

The authors suggest to perform PC measurement as early after venipuncture and ideally while still at body temperature. Do the authors generally recommend POCT-measurement of PC to avoid PTCP?

This and the recommendation to use Plt-F or Plt-O for correct measurement of platelets are not suitable for routine purposes. What is the alternative?

In the reviewers opinion the references 13 and 107 do not justify the statement that the Sysmex XN 9000 and Mindray SF-cube are effectively correcting spurious low platelet counts. To the reviewer´s knowledge there is no convincing study until now.

The reference list needs complete checked: The reference 124 (table 1) is reference 127 in the reference list; Ref. 86, 62, 63 are incomplete (no source, no volume, and no year of issue. 

Table 1 suggests that PTCP is the result of a clinical cause. There are several reports on healthy PTCP cases in the literature! Therefore the column “probable cause” should be omitted. Much more important are the clinical implications including splenectomy.

Author Response

Response to Reviewer’s Comments

Dear Editor,

Coauthors and I very much appreciated the critical and constructive comments on this manuscript by the reviewer. We strongly believe that the comments and suggestions have increased the scientific value of revised manuscript. We have taken them fully into account in revision. We are submitting the corrected manuscript with the suggestion incorporated the manuscript. The manuscript has been revised as per the comments given by the reviewer, and our responses to all the comments are as follows:

Reviewer #1:

  1. Figure 3 : The presentation of numerous data in one figure is not self-explaining and needs modification: The hirudin data do not substantially contribute to the central results or conclusions of the paper and can be summarized in the text and the respective graphs might be omitted. This reduction allows enlarging the remaining graphs and symbols

Response: Thank you very much for your suggestion. However, we believe that the hirudin tube should be included in the presentation as it is increasingly used in clinical laboratories due to its wide use in multiple electrode aggregometry (J Clin Med 2020 Aug 4;9(8):2515. doi: 10.3390/jcm9082515.).Following to your comment, we have added the reason of using hirudin into the manuscript.

“Given the few hirudin-PTCP reports described and its increasing use in clinical laboratories especially for multiple electrode aggregometry [129], the hirudin tube was also studied in this experimental protocol."

  1. How can you prove that the higher PCs as measured by Plt-F and Plt-O result from dissolution of aggregates? It would be therefore of interest to treat a patient´s sample with the respective reagents used for PC measurement and to prepare control smears for aggregate examination

Response: We have taken this reviewer’s comment in full consideration. Since none of the methods was able to completely rule out the presence of aggregates, it is effectively impossible to know whether partial dissociation has taken place for PLT-F/ PLT-O and remains at the hypothesis stage (the presence of clumps on the smears does not allow to know what happened in the analyzer). The hypothesis was suggested based on a recent study that showed that the dissociation effect of optical fluorescence platelet counting on EDTA- PTCP samples was independent of fluorescent dye staining (Transl. Cancer Res., vol. 9, no. 1, pp. 166–172, Jan. 2020, doi: 10.21037/tcr.2019.12.58.). Control smears after treatment with the reagents were not carried out during our experiments. The manuscript has been modified as follows:

"This dissociation effect, independent of manufacturer's fluorescent dye staining, has been postulated in a recent Chinese study [14]. However, sample treatment with hematology or flow cytometry analyzer reagents could not be carried out in our case."

  1. The low PC using K2-EDTA and Plt-I is certainly due to aggregates but also to cell shrinking by reagents and diluents. This assumption explains the increase of PC within the first 60 minutes of storage: K2-EDTA leads to platelet swelling resulting in a proportion of additionally measurable platelets. Therefore it is absolutely necessary to include data of the mean platelet volume (MPV, P-LCR and PDW) of the patient despite probable interferences by the aggregates. Cell shrinking or swelling does not interfere with Plt-F or Plt-O.

These relationships could be substantiated much better if the authors had carried out control measurements with blood from individuals without PTCP.

Response: We appreciate the reviewer’s comment. MPV, P-LCR and PDW  are now provided in the revised manuscript (MPV: 13.3 fL, P-LCR 54.0%, PDW 18.7%). These were obtained at baseline on EDTA-samples. However, no control measurements with blood from individuals without PTCP were carried out during the experiment given the complexity of the protocol. This is now mentioned as a limitation in of the manuscript. The assumption of cell shrinking (which does not interfere with PlT-F and P¨LT-O) was also added in the manuscript to explain the increase of PC within the first 60 minutes in EDTA samples:

“First, control measurements with blood from individuals without PTCP were not carried out. This would have assessed the in vitro cell shrinking or swelling, thus resulting in additionally measurable platelets, observed by the increase of PLT-I within the first hour. This phenomenon does not interfere with PLT-F or PLT-O”

  1. How do the authors explain the low counts in the EDTA-anticoagulated samples without Amikacin when kept at 37 0C and why is this totally different in citrate-anticoagulated blood?

Response: The hypothesis is as follows: amikacin contains a stabilizer which is sodium citrate and the PTCP observed in our patient is partially avoided by sodium citrate/addition of amikacin. Amikacin-free EDTA tubes were therefore the most vulnerable to aggregates. Its addition in the citrated tubes therefore had little or no significant change in platelet levels as sodium fluoride was already present. The influence of temperature, which was discordant between the citrated tubes and EDTA, could not be explained. Different physico-chemical properties or the involvement of different antibodies according to anticoagulants were emitted but not reported in the manuscript.

"In our case, the presence of a stabilizer in the preparation of the aminoglycoside (e.g., sodium citrate) [101] may explain why the addition of amikacin in citrated tubes did not resolve the issue. In contrast, K2-EDTA supplemented with amikacin displayed greater stability of platelet values, thus revealing a possible benefit from dual anticoagulation"

  1. What is the reason that the addition of Amikacin obviously leads to better PCs?

Response: Our patient's PTCP was shown to be rather dependent on the addition of amikacin on EDTA tube. However, as mentioned in the review  amikacin is not always effective in correcting PCs in vitro [13], since its activity is dependent on the type of anticoagulants to which it is added [9]."

  1. The authors postulate that the decreasing PC in the citrated sample is due to calcium release from reversible citrate binding. Is there any proof e.g. smears from parallel experiments?

Response: Thank you for your comment. However no data from parallel experiment were collected and this assumption was based on the current knowledge.

  1. Last not least it is a shortcoming that the interesting results were only obtained from one person in a single experimental approach.
    Some questions could possibly have been answered by replacing hirudin by another anticoagulant e.g. magnesium sulfate.

Response: Indeed, the use of magnesium sulfate would probably have made it possible to remove the interference. Unfortunately, it was not available in our laboratory when the case study was carried out. This is now mentioned as a limitation in of the manuscript:

“Thirdly, other anticoagulant including magnesium sulfate could have been studied but these were not available in our laboratory when the case study was carried out.”

  1. “PTCP patients”? Most candidates are “patients” but not because of PTCP.
    “Individuals with PTCP” is more correct. PTCP is a laboratory in vitro phenomenon.

Response: Thank you very much for your comment. This is has been modified in the revised version of manuscript as follows : “PTCP patients” to “individuals with PTCP”.

  1. The authors suggest to perform PC measurement as early after venipuncture and ideally while still at body temperature. Do the authors generally recommend POCT-measurement of PC to avoid PTCP?

Response: POCT should not be recommended except in specific situations where the interference cannot be removed and after testing other alternatives. Rapid analysis is suggested at the end of the algorithm, when multi-anticoagulant PTCP is encountered.

  1. This and the recommendation to use Plt-F or Plt-O for correct measurement of platelets are not suitable for routine purposes. What is the alternative?

Response: In our laboratory using Sysmex XN series analyzers, fluorescence and optical platelet counting could be carried out routinely. As described in the algorithm, when these technologies are not available, smear inspection and the use of alternative anticoagulants is desired.

  1. In the reviewers opinion the references 13 and 107 do not justify the statement that the Sysmex XN 9000 and Mindray SF-cube are effectively correcting spurious low platelet counts. To the reviewer´s knowledge there is no convincing study until now.

Response: Reference numbers were incorrectly assigned. They have been rechecked for the entire review. However, the two correctly cited authors demonstrated efficacy in restoring platelet counts using fluorescence counting in 6 and 23 individuals with pseudo-thrombocytopenia to EDTA, respectively (Transl. Cancer Res., vol. 9, no. 1, pp. 166–172, Jan. 2020, doi: 10.21037/tcr.2019.12.58. & Clin. Chim. Acta, vol. 502, pp. 99–101, Mar. 2020, doi: 10.1016/j.cca.2019.12.012). We have now mentioned that additional studies on a large number of patients are still required to demonstrate that Sysmex XN 9000 and Mindray SF-cube are effectively correcting spurious low platelet counts.

  1. The reference list needs complete checked: The reference 124 (table 1) is reference 127 in the reference list; Ref. 86, 62, 63 are incomplete (no source, no volume, and no year of issue. 

Response: Thanks for your comment. The reference list was checked and the incomplete reference were modified in the revised manuscript.

  1. Table 1 suggests that PTCP is the result of a clinical cause. There are several reports on healthy PTCP cases in the literature! Therefore the column “probable cause” should be omitted. Much more important are the clinical implications including splenectomy.

Response: Following to your comment, the column heading has been changed from “probable cause” to "medical condition”" so that the reader can understand the significance of the clinical impact of PTCP in its context. Medical conditions have therefore been added for Kohlschein et al., 2015 and Greinacher et al., 2016 .

Reviewer 2 Report

P1 - Abstract:

Please elaborate the clinical consequences and potential life-threatening events in patients with PTCP. An example would illustrate which severe complications are meant. 

P1 - Abstract; Introduction Part (i); Page 3 Line 3:  

Please clarify the amikacin supplementation. Readers may ask, if amikacin is given as antibiotic or used as an additive.

P 7 Line 18:

Please state the meaning of autoimmune (i.e. antibodies, disease)

P 12 Line 9-11:

...last recommendations of Groupe Francophone d’Hématologie Cellulaire (GFHC) for thrombocytopenia.

Please provide a citation

P 14 Conclusions Part:

It is associated or not with some disorders ...

The meaning of this sentence is not clear. Please elaborate.

Author Response

Reviewer #2:

  1. P1 - Abstract:Please elaborate the clinical consequences and potential life-threatening events in patients with PTCP. An example would illustrate which severe complications are meant. 

Response: Thank you for your comments. Abstract section has been revised considering your valuable suggestion: “Clinical consequences with potential life-threatening events (e.g. unnecessary platelet transfusion, wrong treatment including splenectomy or corticosteroids) are still observed when PTCP is not readily detected”.

  1. P1 - Abstract; Introduction Part (i); Page 3 Line 3:  Please clarify the amikacin supplementation. Readers may ask, if amikacin is given as antibiotic or used as an additive.

Response: Thank you for your comment. “in vitro” term has been added to the “amikacin supplementation” in order to specify the additive aspect of this one.

  1. P 7 Line 18: Please state the meaning of autoimmune (i.e. antibodies, disease)

Response: Thank you for your comment. This sentence has been restructured to make it clearer for the reader: “The commonly accepted hypothesis entails antibody production due to cross-reactivity, as recently described in two COVID-19 patients during viral immunization [2,3], or in autoimmune antibodies as described here. “ The term “disease” has not been mentioned as no clinical signs of autoimmune disease have been reported in our patient. Only laboratory investigations proved high antinuclear antibodies titer and platelet-bound GPIIbIIIa antibodies.

  1. P 12 Line 9-11:...last recommendations of Groupe Francophone d’Hématologie Cellulaire (GFHC) for thrombocytopenia. Please provide 0a citation

Response: Thank you for your comment. A citation has been provided.

  1. P 14 Conclusions Part:“It is associated or not with some disorders ...”The meaning of this sentence is not clear. Please elaborate.

Response : Thank you for your comment. This sentence has been reworded and clarified :

"It seems that no particular disease may be specially associated with PTCP, so that additional investigations could be avoided in the absence of suggestive clinical signs. (…)Observing PTCP in individuals does not increase the risk of developing future disorder. Conversely, the incidence of PTCP is higher in patients with identified risk factors, including male sex, age over 50 years, underlying diseases or therapy with drugs such as LMWH."

Round 2

Reviewer 1 Report

The submitted revision has taken into account most of the reviewers annotations and suggestions. Nevertheless some critical points remain still open.

Major points:

Figure 2:

I still have a problem with the control experiments:
The control shows aggregates independent from the anticoagulant used, but at which temperature? I guess at RT. The smears were performed at baseline, directly after blood sampling?
The addition of Amikacin does not show any difference at RT, but at 370C, mainly in the citrate anticoagulated sample. The correct control would have been a control at 370C without Amikacin to exclude a general effect of 370C on platelet aggregates. 
If Amikacin at 370C really leads to less aggregates I would expect more non-aggregated platelets in the smear.

Figure :3

What is the difference between "Control, RT" and 
"No Amikacin, RT"? Where are data of "Control, 370C? 
I suppose that there were no platelets at all in the EDTA-anticoagulated samples without Amikacin, otherwise I cannot explain that no platelets were measurable, independent from the method. This might be a performance or handling error. Do you have had a controle smear of this approach? Did you measure platelets in this sample by FCM?

As suggested you now added data for MPV but I do not believe that they   remained stable over the whole period of measurement. It would have been  interesting to compare the initial measurement and that after e.g. 60 min. Was there no difference of MPV between EDTA-anticoagulated and citrate anticoagulated platelets? If you measured PCs by the Sysmex XN you should have all these data.

Minor points:

p. 10, line 325: The Chinese study needs confirmation by a controlled study, therefore I suggest to use the term "assumed" instead of "postulated". 

p. 16, line 494-496: The second part of the sentence  "...., so that additional investigations could be avoided in the absence of suggestive clinical signs" can be omitted.

Author Response

Dear editor,

Thank you again for these valuable suggestions. You will find answers to the questions and changes to the manuscript below.

Major points:

  1. Figure 2:

-     1.1 The control shows aggregates independent from the anticoagulant used, but at which temperature? I guess at RT.

Response: Yes, the control tubes for EDTA, citrate and hirudin were at room temperature as mentioned in the manuscript: “A patient control sample without amikacin or diluent was maintained at room temperature (RT) for each set.”

-     1.2 The smears were performed at baseline, directly after blood sampling?

Response: Yes, the smears were carried out within 30 seconds of blood collection as specified in Figure 1. This time included gently mixing, opening the cap, adding amikacin or manufacturer’s diluent, closing the cap, another mixing, opening the cap again, pipetting, and spreading whole blood on the slide. This procedure was rehearsed several times before being performed. One person realized the venipuncture while the two others handled the tubes and smears.

-     1.3 The addition of Amikacin does not show any difference at RT, but at 370C, mainly in the citrate anticoagulated sample. The correct control would have been a control at 370C without Amikacin to exclude a general effect of 370C on platelet aggregates. 

Response:  Indeed, given the results obtained, it would have been interesting to obtain such a control. However, we decided to consider a single control tube as a normal routine tube, without adding any variable. We added this as a limitation and the manuscript has been modified as follows:

This single person experimental approach had some limitations. First, control measurements with blood from individuals without PTCP were not carried out. This would have assessed the in vitro cell shrinking or swelling, thus resulting in additionally measurable platelets, observed by the increase of PLT-I within the first hour. This phenomenon does not interfere with PLT-F or PLT-O. Secondly, no control tube at 37°C was obtained therefore a general effect of temperature  on platelet clumps could not be excluded. Thirdly, no data was available to explain the decreasing PC in citrated sample, though the hypothesis of calcium release from reversible citrate binding could be suggested. Fourthly, other anticoagulants including magnesium sulfate could have been studied but these were not available in our laboratory when the case study was carried out.

-     1.4 If Amikacin at 370C really leads to less aggregates I would expect more non-aggregated platelets in the smear.

Response: Although clumps appeared to be less platelet-rich with amikacin, it remained difficult to establish a relationship between the platelet count obtained by the analyzer and the degree of clumping. In fact, some clumps were at/below the limit of the number of platelets required to be considered as a clump by the GFHC (5 platelets): “platelet clump was defined as a minimum of five attached platelets [4].

  1. Figure 3 :

-     2.1 What is the difference between "Control, RT" and 
"No Amikacin, RT"? Where are data of "Control, 37C? 

Response: As mentioned in the manuscript, “Amikacin-free specimens were diluted with manufacturer’s diluent, for achieving the same dilution as in tubes containing amikacin; the dilution effect of sodium citrate, amikacin or diluent were calculated.” The control RT is therefore a normal sample without additive and the "No amikacin, RT" is a tube containing manufacturer's diluent. There was no control tube at 37°C. As mentioned in the manuscript: “A patient control sample without amikacin or diluent was maintained at room temperature (RT) for each set.” but not at 37°C. This has been added in the limitations (cfr 1.3).

-     2.2 I suppose that there were no platelets at all in the EDTA-anticoagulated samples without Amikacin, otherwise I cannot explain that no platelets were measurable, independent from the method. This might be a performance or handling error. Do you have had a control smear of this approach? Did you measure platelets in this sample by FCM?

Response: The blood smear of the edta-anticoagulated sample without amikacin is not shown in Figure 1, but the latter also showed numerous clumps on the smear. No macroscopic clot was observed. The PLT IP message delivered 2 flags (“PLT Abn Distribution” and  “Thrombocytopenia”). Flow cytometry was carried out on both K2-EDTA (control tube) and K2-EDTA supplemented with amikacin but not on the K2-EDTA amikacin-free specimen (which was added with manufacturer’s diluent). PC with FCM was 145.109/L and 142.109/L, respectively, while the latter tubes also showed many clumps on the blood smear.  This has been added in the manuscript in the Figure 2 legend and later in the manuscript:

Figure 2: Platelet clumps observed on blood smears from samples obtained at baseline according to different anticoagulants tubes and conditions. Clumps were observed until the last observation at 240 minutes. Although not illustrated, amikacin-free specimens also displayed several clumps.

(…)Maintenance of EDTA tubes at 37°C rather than at RT had negative impact on PC, with all analytical techniques used. This was particularly observed for amikacin-free specimens  at 37°C, with many clumps observed on the blood smear. Hirudin specimens displayed the worst stability over time, in all conditions.  

-     2.3.As suggested you now added data for MPV but I do not believe that they   remained stable over the whole period of measurement. It would have been  interesting to compare the initial measurement and that after e.g. 60 min. Was there no difference of MPV between EDTA-anticoagulated and citrate anticoagulated platelets? If you measured PCs by the Sysmex XN you should have all these data.

Response:  Indeed, we have some of these data because the Sysmex analyzer could not render the MPV on all the measurements. No explanation were provided and MPV, PDW, P-LCR and PCT were not given by the analyzer without a specific reason or special flags.  The MPV on the baseline citrate control tube was 11.7 fL. There was therefore a difference with the EDTA tube. The MPV on the EDTA control tube  at 60 minutes was 15.1 fL, which is consistent with the platelet swelling due to edta. The MPV on the citrate tube at 60 minutes was not obtained but was 12.7 fL at 180 and 240 minutes. The manuscript has been modified in the limitations as follows :

“This would have assessed the in vitro cell shrinking or swelling, thus resulting in additionally measurable platelets, observed in the EDTA control tube by the increase of PLT-I and MPV from 13.3 to 15.1 fL within the first hour. This phenomenon does not interfere with PLT-F or PLT-O.

Minor points:

- p. 10, line 325: The Chinese study needs confirmation by a controlled study, therefore I suggest to use the term "assumed" instead of "postulated". 

Response: .Thank you for this valuable suggestion. The sentence has been modified: “This dissociation effect, independent of manufacturer's fluorescent dye staining, has been assumed in a recent Chinese study [14]. »

- p. 16, line 494-496: The second part of the sentence  "...., so that additional investigations could be avoided in the absence of suggestive clinical signs" can be omitted.

Response: Thank you for you comment. The sentence has been reduced : “It seems that no particular disease may be specially associated with PTCP”.